# Synergy between serum amyloid A and secretory phospholipase A$_2$

**Shobini Jayaraman[1]\*, Marcus Fändrich[2], Olga Gursky[1,3]**

[1]Department of Physiology and Biophysics, Boston University School of Medicine, Boston, United States; [2]Institute of Protein Biochemistry, Ulm University, Ulm, Germany; [3]Amyloidosis Treatment and Research Center, Boston University School of Medicine, Boston, United States

**Abstract** Serum amyloid A (SAA) is an evolutionarily conserved enigmatic biomarker of inflammation. In acute inflammation, SAA plasma levels increase ~1,000 fold, suggesting that this protein family has a vital beneficial role. SAA increases simultaneously with secretory phospholipase A$_2$ (sPLA$_2$), compelling us to determine how SAA influences sPLA$_2$ hydrolysis of lipoproteins. SAA solubilized phospholipid bilayers to form lipoproteins that provided substrates for sPLA$_2$. Moreover, SAA sequestered free fatty acids and lysophospholipids to form stable proteolysis-resistant complexes. Unlike albumin, SAA effectively removed free fatty acids under acidic conditions, which characterize inflammation sites. Therefore, SAA solubilized lipid bilayers to generate substrates for sPLA$_2$ and removed its bioactive products. Consequently, SAA and sPLA$_2$ can act synergistically to remove cellular membrane debris from injured sites, which is a prerequisite for tissue healing. We postulate that the removal of lipids and their degradation products constitutes a vital primordial role of SAA in innate immunity; this role remains to be tested in vivo.

DOI: https://doi.org/10.7554/eLife.46630.001

**\*For correspondence:** shobini@bu.edu

## Introduction

The serum amyloid A (SAA) family consists of 12-kDa proteins that have been highly evolutionarily conserved at least since the Cambrian period, from sea cucumber to human (*Uhlar et al., 1994*; *Sun et al., 2016*). SAA is an enigmatic biomarker of inflammation that is better known as a protein precursor of systemic amyloid A (AA) amyloidosis, a life-threatening complication of chronic inflammation, than for its beneficial action (*Westermark et al., 2015*; *Papa and Lachmann, 2018*). Inducible human SAA is produced mainly by the liver under the control of pro-inflammatory cytokines, is secreted into blood, and binds its major plasma carrier, high-density lipoprotein (HDL) (*Benditt and Eriksen, 1977*). SAA is also secreted locally at inflammation sites and is implicated in cytokine production and immune cell recruitment to these sites (*De Buck et al., 2016*; *Eklund et al., 2012*; *Ye and Sun, 2015*). During the acute-phase response, which is a complex systemic response to severe inflammation, infection or injury (*Gabay and Kushner, 1999*), human SAA isoforms 1, 2 and 3 are upregulated (*Uhlar and Whitehead, 1999*), while isoform 4 is constitutively expressed at much lower levels (reviewed in *Sun et al., 2016* and in *De Buck et al., 2016*). Plasma levels of inducible SAA are elevated in infections such as tuberculosis, in autoimmune disorders such as rheumatoid arthritis, lupus, and Crohn's disease, and in certain cancers (*De Buck et al., 2016*; *Eklund et al., 2012*; *Ye and Sun, 2015*; *Sack, 2018*). Although chronically elevated SAA is deleterious as a protein precursor of amyloidosis and as a causal risk factor for atherosclerosis (*Eklund et al., 2012*; *Getz et al., 2016*; *Thompson et al., 2018*), the beneficial action of SAA is less clear. In fact, SAA has been reported to be pro- or anti-inflammatory in various studies, and its functions in acute and

**eLife digest** Cell boundaries are made up of fatty substances known as lipids. When cells get severely damaged, their lipid membranes break apart. These broken fragments of membrane become highly toxic, and must be removed as soon as possible to allow the tissue to heal. A small protein called serum amyloid A, SAA for short, was recently proposed to play a pivotal role in this process. In humans, SAA levels in the blood rapidly spike to over a thousand times their normal level following inflammation, injury or infection. Combined with the fact SAA has been conserved for over 500 million years, this suggests that SAA must be important for survival. But, it is not entirely clear how this protein works.

One clue for how SAA works is its relationship to another ancient protein called secretory phospholipase $A_2$. This protein, also known as $sPLA_2$, is part of a big family of enzymes that break down lipids in the cell membrane. Notably, $sPLA_2$ levels rise at the same time and place as SAA during inflammation. This led Jayaraman et al. to ask whether SAA and $sPLA_2$ might be working together to clean up the cell membrane debris.

To find out, Jayaraman et al. mixed mouse SAA with vesicles of membrane lipids, and then added $sPLA_2$. This revealed that SAA reshapes the lipid membrane into smaller 'nanoparticles' with tightly curved surfaces that are easier for $sPLA_2$ to break down. As the $sPLA_2$ breaks up these particles, SAA then gathers up and gets rid of the leftover toxic fragments. This suggests that SAA has two roles: helping $sPLA_2$ break down the membrane, and removing any toxic debris.

Clearing debris after injury is essential for proper healing. So, understanding how it works is crucial to find new ways to treat inflammation. Further work to understand SAA and $sPLA_2$ could improve our understanding of how to treat acute and chronic inflammation and its life-threatening complications.

DOI: https://doi.org/10.7554/eLife.46630.002

chronic inflammation remain enigmatic (reviewed in *Eklund et al., 2012*, *Ye and Sun, 2015*, *Sack, 2018*, and *Kisilevsky and Manley, 2012*).

Remarkably, in acute inflammation, during infection, after injury or following surgery, plasma levels of SAA increase swiftly more than 1,000-fold, reaching up to 3 mg/ml in 24–48 hr, and then the levels drop (*Sun et al., 2016*; *De Buck et al., 2016*; *Uhlar and Whitehead, 1999*; *Sack, 2018*). The advantage for survival of this dramatic but transient increase is unclear. However, high sequence conservation in this ancient protein family (*Uhlar et al., 1994*; *Sun et al., 2016*) and a major and rapid commitment of liver and local tissues to SAA biosynthesis suggest that SAA is vital for survival.

One potential beneficial role of SAA is its ability to mobilize HDL cholesterol for cell repair. In the acute-phase response, SAA becomes a major HDL protein that can reroute the transport of HDL cholesterol by interacting with several cellular scavenger receptors that bind SAA-modified HDL (reviewed in *Kisilevsky and Manley, 2012*). However, HDL undergoes additional modifications during the acute-phase response (*Jahangiri, 2010*; *Tall and Yvan-Charvet, 2015*), and the role of SAA in the homeostasis of these modified HDLs can be relatively minor (*de Beer et al., 2013*). Moreover, rerouting HDL cholesterol transport cannot explain rapid and massive secretion of SAA over a period of hours following the onset of acute inflammation in various organisms, including those lacking HDL. Hence, the key primordial function of SAA must be different from HDL homeostasis.

Although most circulating SAA is bound to HDL, like other HDL proteins, SAA is an exchangeable apolipoprotein that can transiently dissociate in a labile 'free' form (*Wilson et al., 2018*). Free SAA can bind a range of other apolar ligands, including cholesterol (*Liang et al., 1996*), retinol (*Derebe et al., 2014*), phospholipids, lysophospholipids, and free fatty acids (FFA) (*Takase et al., 2014*; *Jayaraman et al., 2015*; *Frame and Gursky, 2016*; *Tanaka et al., 2017*; *Jayaraman et al., 2017a*; *Frame et al., 2017*; *Jayaraman et al., 2018*). Our in vitro studies showed that SAA binds various phospholipid vesicles and spontaneously solubilizes them to form HDL-sized particles de novo (*Frame et al., 2017*; *Jayaraman et al., 2018*). We proposed that this ability hinges upon the binding of diverse apolar ligands at a large concave apolar face of the SAA molecule (*Frame and Gursky, 2016*). This face, formed by two amphipathic α-helices, was observed in the atomic-resolution x-ray crystal structures of human SAA1.1 and murine SAA3 (*Derebe et al., 2014*;

*Lu et al., 2014*). The shape of this apolar face, whose key features are conserved in the SAA family (*Frame and Gursky, 2016*), helps to explain the preferential binding of SAA to highly curved apolar surfaces, a property that is essential for HDL binding and lipid sequestration (*Frame and Gursky, 2016*; *Frame et al., 2017*). These findings compelled us to propose that SAA's ability to solubilize phospholipid bilayers and to form lipoprotein nanoparticles de novo reflects the primordial role of this Cambrian protein in the removal of cell membrane debris from injured sites, a function that pre-dates SAA binding to HDL (*Frame et al., 2017*).

Here, we consider a functional link between SAA and another ancient lipophilic plasma protein, phospholipase $A_2$ ($PLA_2$). $PLA_2$ is a superfamily of diverse enzymes that hydrolyze phospholipids in the sn2 position (*Burke and Dennis, 2009*). The reaction products, lysophospholipids and FFA, are bioactive lipids that are precursors of signaling molecules in many vital processes (*Burke and Dennis, 2009*). Secretory $PLA_2$ ($sPLA_2$) is a family of pro-inflammatory enzymes that are involved in the immune response (*Boyanovsky and Webb, 2009*; *Murakami et al., 2016*), which is especially relevant to SAA. For example, $sPLA_2$ group-IIa ($sPLA_2$-IIa) is an antimicrobial acute-phase reactant whose concentration in plasma and at inflammation sites can increase several hundred-fold simultaneously with that of SAA (*Pruzanski et al., 1993*). Notably, $sPLA_2$ is co-expressed with SAA and is induced by the same group of pro-inflammatory cytokines (*Vadas et al., 1993*). Moreover, SAA stimulates smooth muscle cells to express $sPLA_2$-IIa (*Sullivan et al., 2010*). Clinical studies have reported a direct link between the plasma levels of SAA and the enhanced activity of $sPLA_2$ during the early stages of inflammation, whereas in vitro studies have shown that SAA enhances the remodeling of $sPLA_2$-induced lipoproteins via an unknown mechanism (*Pruzanski et al., 1995*; *Pruzanski et al., 1998*). Furthermore, $sPLA_2$ hydrolyzes highly curved micelle-like surfaces in lipoproteins such as HDL (diameter 8–12 nm), but not intact planar bilayers (*Høyrup et al., 2004*; *Halperin and Mouritsen, 2005*), whereas SAA preferentially binds to such highly curved surfaces or forms them de novo by solubilizing lipid bilayers (*Jayaraman et al., 2018*; *Lu et al., 2014*). Taken together, these findings compel us to postulate not only a spatiotemporal overlap between SAA and $sPLA_2$ at inflammation sites in vivo, but also their potential synergy in lipid clearance (*Jayaraman et al., 2016*). This study explores this synergy and its mechanism.

## Results

### SAA augments the lipolysis of both model and plasma lipoproteins by $sPLA_2$

The murine and human SAA isoform 1.1 (mSAA1.1 and hSAA1.1) proteins used in this study are major isoforms that bind HDL and form amyloid in vivo (*Westermark et al., 2015*). Recombinant mSAA1.1 (hereafter termed SAA for brevity) was used in most experiments. To determine whether and how SAA influences phospholipid hydrolysis by $sPLA_2$, model and plasma lipoproteins that differed in size and composition were used as substrates for either $sPLA_2$ group-III ($sPLA_2$-III), which preferentially hydrolyses phosphatidylcholine (PC), or $sPLA_2$-IIa, which preferentially hydrolyses phosphatidylethanolamine but also acts on PC (*Burke and Dennis, 2009*; *Boyanovsky and Webb, 2009*). Unless otherwise stated, both $sPLA_2$-III and $sPLA_2$-IIa enzymes are collectively termed $sPLA_2$.

We first probed how the lipid surface curvature imposed by SAA influences the $sPLA_2$ reaction. SAA was incubated with multilamellar vesicles (MLV) of a model phospholipid palmitoyl-oleoyl-PC (POPC). The results presented in *Figure 1* show that SAA solubilized POPC MLV (diameter circa 200 nm) to form smaller SAA-POPC complexes (~8 nm). The time course of this microsolubilization was monitored by turbidity, and its products were observed by non-denaturing PAGE (*Figure 1A,B*). Increasing the SAA to POPC molar ratio from 1:100 to 1:10 increased the rate of solubilization (*Figure 1A*) and resulted in the generation of slightly smaller SAA-POPC particles; for all ratios explored, the particle size was 7.5–8.5 nm (*Figure 1B*). These small particles, along with POPC MLV, were used as substrates for $sPLA_2$, whose enzymatic activity was assessed by measuring free fatty acid products. In the absence of $sPLA_2$, no significant hydrolysis of SAA-POPC particles was detected, but in the presence of $sPLA_2$, extensive hydrolysis of the SAA-POPC particles was observed (*Figure 1C*). By contrast, MLV were not hydrolyzed by $sPLA_2$: the levels of FFA were below the detection limit of our assay, and thin-layer chromatography showed only the presence of PC (*Figure 1D*). Consequently, SAA readily solubilizes model phospholipid bilayers such as POPC MLV

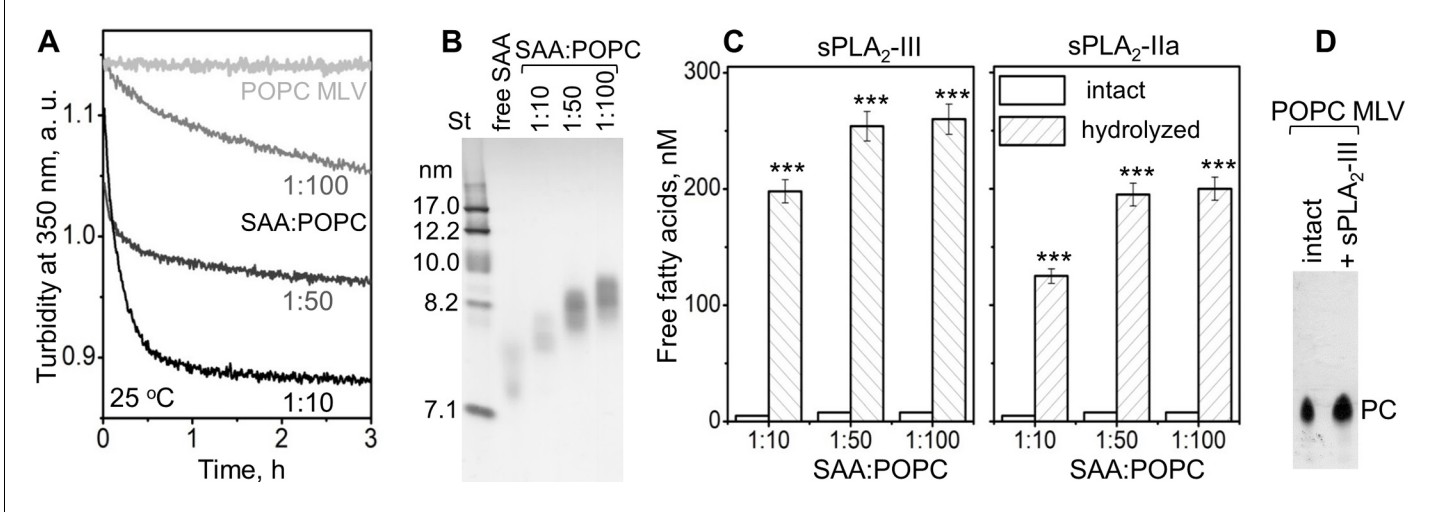

**Figure 1.** SAA remodels phospholipid bilayers into small particles that form substrates for sPLA$_2$. (**A**) POPC MLV (~200 nm) were incubated with SAA at 25°C, and the time course of MLV remodeling into smaller particles was monitored by turbidity at 350 nm. Protein:lipid molar ratios were 1:10, 1:50 and 1:100, as indicated. Protein-free POPC MLV were used as a control. The data for POPC MLV alone (light gray) closely superimposed similar data using POPC MLV with either apoA-I or apoA-II at a 1:100 protein:lipid weight ratio (not shown to avoid overlap). (**B**) Non-denaturing PAGE of the POPC mixtures with SAA, performed after 6 hr of incubation, shows the formation of SAA-POPC complexes. SAA:POPC molar ratios are indicated; lipid-free SAA is shown for comparison. Gels in this and other figures were stained with Denville Blue protein stain. (**C**) SAA-POPC particles shown in panel (**B**) were isolated by size-exclusion chromatography (SEC) and used as substrates for sPLA$_2$-III (left) or sPLA$_2$-IIa (right). Free fatty acids produced per nm of lipid are shown as average values of three independent measurements with standard errors of mean. The protein:lipid molar ratios in the initial incubation mixtures are indicated; sPLA$_2$-free particles were used as controls. FFA produced in the presence and in the absence of sPLA$_2$ were compared using the t-test; ***p<0.001. In SAA-POPC particles, approximately 70% of the total lipid was hydrolyzed by sPLA$_2$-III and 60% by sPLA$_2$-IIa. (**D**) Thin-layer chromatography analysis of POPC MLV before (intact) or after their incubation with sPLA$_2$-III (+ sPLA$_2$-III). The PC band is indicated; the absence of the lysoPC band underneath the PC band indicates the absence of significant hydrolysis.

DOI: https://doi.org/10.7554/eLife.46630.003

The following source data is available for figure 1:

**Source data 1.** Free fatty acid analysis of SAA:POPC complexes hydrolysed by sPLA2.

DOI: https://doi.org/10.7554/eLife.46630.004

and converts them into small HDL-size particles that provide excellent substrates for sPLA$_2$. This ability distinguishes SAA from other major HDL proteins, such as apoA-I and apoA-II, which cannot spontaneously solubilize POPC MLV (*Figure 1A*).

Next, we tested the effects of SAA on the lipolysis of plasma HDL by sPLA$_2$. Human HDL that contained various amounts of bound SAA (up to 27% of the total protein mass), termed SAA-HDL, were prepared by incubation of HDL with SAA using 1:1 or 4:1 protein weight ratio of exogenous SAA to endogenous apoA-I as described in the 'Materials and methods' (*Figure 2—figure supplement 1A–C*). The lipoprotein fraction containing only HDL-bound proteins was isolated by size-exclusion chromatography (SEC) from the total incubation mixture (marked SEC Fr and total in *Figure 2—figure supplement 1B*), and was hydrolyzed by sPLA$_2$. A progressive increase in activity with an increasing amount of bound SAA was observed (*Figure 2A,B*). As the particle curvature was similar in these experiments, this increased activity must have stemmed from the presence of bound SAA.

To determine whether unbound (free) SAA also enhanced the enzymatic activity of sPLA$_2$, we used sPLA$_2$ to hydrolyze total incubation mixtures that contained HDL-bound and free proteins (*Figure 2—figure supplement 1A*). For the same amount of SAA, the enhancementof lipolytic activity was comparable in the presence of HDL-bound SAA or a mixture of HDL-bound and free SAA (compare 4:1 SAA-HDL Sec Fr in *Figure 2A,B* with SAA-HDL total in *Figure 2C,D*). To directly probe the role of free SAA, we incubated it with normal human LDL that does not bind SAA; the mixture (SAA +LDL, *Figure 2—figure supplement 1D*) was hydrolyzed with sPLA$_2$. Increased lipolysis was

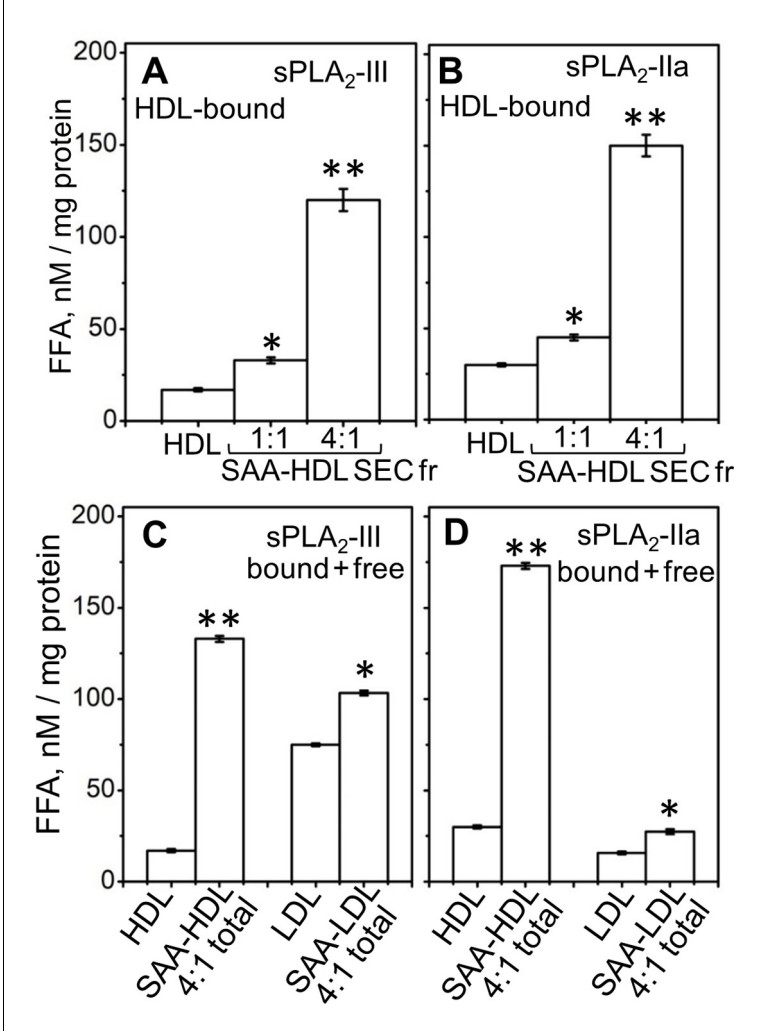

**Figure 2.** Effect of SAA on the lipolysis of HDL and LDL by sPLA$_2$-III or sPLA$_2$-IIa. (A, B) FFA were generated upon lipolysis of either native HDL or SAA-HDL. For SAA-HDL, the SEC fraction containing HDL-bound proteins (Sec Fr) was isolated from the total incubation mixtures, which contained 1:1 or 4:1 SAA:apoA-I molar ratios as indicated. (C, D) FFA were generated upon lipolysis of HDL, LDL and their total incubation mixtures with SAA. Protein weight ratios were 1:4 SAA to apoA-I in HDL or 1:4 SAA to apoB in LDL. The amounts of FFA produced per mg of apoA-I (in HDL) or apoB (in LDL) are shown as the means of three independent measurements with the standard deviations of these means. FFA generated in the presence and in the absence of SAA were compared using one-way ANOVA; *, p<0.05; **, p<0.01. A characterization of protein-containing complexes formed upon incubation of SAA with human plasma lipoproteins is shown in *Figure 2—figure supplement 1*.

DOI: https://doi.org/10.7554/eLife.46630.005

The following source data and figure supplement are available for figure 2:

**Source data 1.** Free fatty analysis of HDL and LDL hydrolysed by sPLA2.
DOI: https://doi.org/10.7554/eLife.46630.007

**Figure supplement 1.** Characterization of protein-containing complexes formed upon incubation of SAA with human plasma lipoproteins.
DOI: https://doi.org/10.7554/eLife.46630.006

observed upon addition of free SAA to LDL (*Figure 2C,D*). Consequently, free SAA augmented the lipolysis by sPLA$_2$ of various model and plasma lipoproteins.

In summary, the results in *Figure 1* and *Figure 2—figure supplement 1D* show that SAA enhances the hydrolysis by sPLA$_2$-III or sPLA$_2$-IIa of diverse substrates, including SAA-POPC complexes, plasma HDL, and plasma LDL. This enhancement reflects the possibilities that: i) SAA not only binds

to phospholipid bilayers but also remodels them into smaller highly curved HDL-size particles (~8 nm) that are readily hydrolyzed by sPLA$_2$ (*Figure 1D,E*), and ii) SAA augments the action of sPLA$_2$ in a manner that does not involve SAA binding to the substrate, as evident from the SAA-induced enhancement of LDL lipolysis (*Figure 2C,D*). As we and others have shown that SAA binds FFA and lysoPC in vitro (*Tanaka et al., 2017*; *Jayaraman et al., 2018*), the latter effect could stem from interactions of SAA with the products of sPLA$_2$. This idea was tested as described below.

## Lipid hydrolysis by sPLA$_2$ in the presence of SAA generates 7–7.5 nm species

The ~8 nm SAA-POPC complexes formed upon spontaneous solubilization of MLV using 1:10 to 1:100 protein:lipid molar ratioswere nearly invariant in size (*Figure 1C*). By contrast, SAA incubated with small uninlamellar vesicles (SUV) of POPC formed particles that increased in size as protein:lipid ratio decreased (*Frame et al., 2017*). Therefore, in the current study, we used POPC SUV to test the effect of particle size on lipolysis by sPLA$_2$. To test whether lipoprotein hydrolysis by sPLA$_2$ involved changes in particle size in the absence and in the presence of SAA, we used non-denaturing PAGE to analyze model and plasma HDL before and after the lipolysis. First, SAA-POPC complexes that varied in size from about 8 nm to 22 nm were prepared by incubating SAA with POPC SUV at protein to lipid molar ratios ranging from 1:1 to 1:100. These incubation mixtures were hydrolyzed by sPLA$_2$ for 3 hr at 37°C as described in 'Materials and methods'. Non-denaturing PAGE showed that all parent particles were remodeled by sPLA$_2$ into species that migrated at 7–7.5 nm (*Figure 3— figure supplement 1A*).

Next, we performed similar studies using plasma HDLs that were either native or enriched with exogenous SAA (SAA-HDL) as described in 'Materials and methods'; these HDLs were used as substrates for sPLA$_2$. Before hydrolysis, both native HDL and SAA-HDL ranged in size from about 8.5 to 12 nm. Hydrolysis of native HDL by sPLA$_2$ caused little change in the particle size distribution and no significant protein release (*Figure 3—figure supplement 1B*). By contrast, hydrolysis of SAA-HDL led to lipoprotein remodeling into two major protein-containing species, of 10–12 nm and 7–7.5 nm in size (*Figure 3—figure supplement 1B*). Together, the results in *Figure 3—figure supplement 1* suggest that sPLA$_2$ hydrolysis of SAA-POPC particles and SAA-HDL, but not of native HDL, leads to a release of protein-containing species that have a hydrodynamic size of 7–7.5 nm.

We tested whether similar species were formed upon direct interaction of SAA with hydrolyzed phospholipids. First, POPC SUV were incubated with sPLA$_2$ for 3 hr at 37°C as described in 'Materials and methods', leading to the hydrolysis of 40–50% of the POPC. Next, the hydrolyzed samples were incubated at 25°C for 6 hr with free SAA using protein to PC molar ratios ranging from 1:1 to 1:100. Non-denaturing PAGE showed that the particle size distribution varied depending upon the initial protein to lipid ratio, yet at all ratios, the major protein-containing species were observed at 7–7.5 nm (*Figure 3A*, right panel). A strikingly similar migration pattern was observed for the SAA-POPC complexes that were formed using 1:1 to 1:100 protein to PC ratios and then hydrolyzed (*Figure 3—figure supplement 1A*). Therefore, regardless of the order of the events (binding to SAA and hydrolysis by sPLA$_2$), SAA formed 7–7.5 nm complexes with the hydrolytic products of POPC, suggesting that these complexes represented stable, kinetically accessible species.

Incubation of free SAA with sPLA$_2$-treated HDL (*Figure 3B*) or LDL (*Figure 3C*) also led to the formation of distinct 7–7.5 nm species. Such species were released from parent lipoproteins only in the presence of both SAA and hydrolyzed phospholipids, and were detected in model systems and in plasma lipoproteins (*Figure 3A–C*). We reasoned that all of these 7–7.5 nm species could represent stable complexes of SAA with the products of phospholipid hydrolysis, and tested this idea in the following experiments.

## SAA sequesters the FFA and lysoPC produced by sPLA$_2$ in model and plasma lipoproteins

To determine the properties of the 7–7.5 nm species formed in the presence of SAA and hydrolyzed phospholipids, these species were isolated by density gradient centrifugation. SAA-POPC particles of 8 nm in size, as well as SAA-containing samples of HDL or LDL — containing a 4:1 protein weight ratio of SAA to either apoA-I (which is the major HDL protein) or apoB (which is the major LDL

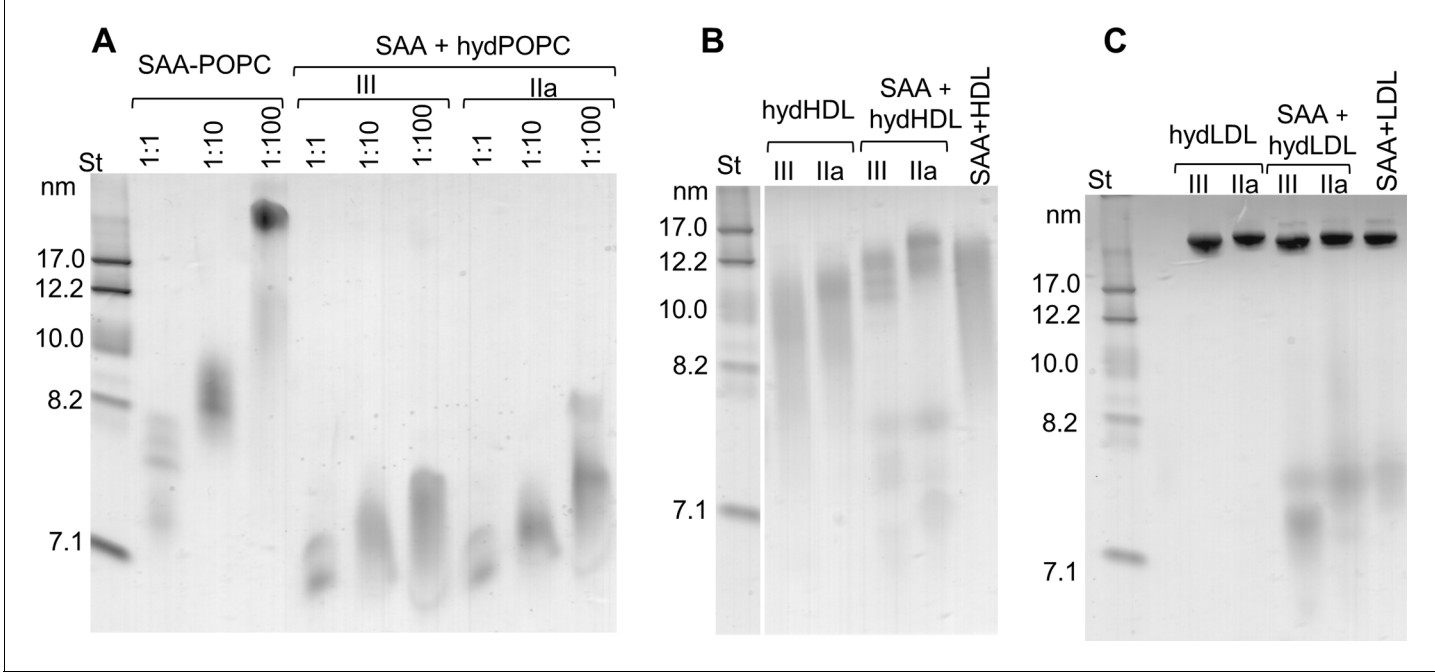

**Figure 3.** SAA forms complexes with hydrolyzed model or plasma lipids. (**A**) SAA was incubated with either unmodified POPC to form SAA-POPC complexes or with hydrolyzed POPC to form SAA + hydPOPC complexes (see 'Materials and methods' for details). Protein:POPC molar ratios were 1:1, 1:10 or 1:100 as indicated; sPLA$_2$-III or sPLA$_2$-IIa was used as indicated. (**B, C**) Human plasma lipoproteins including HDL (**B**) and LDL (**C**) were hydrolyzed with sPLA$_2$ group-III or -IIa to form hydHDL or hydLDL, respectively. In samples marked SAA + hydLDL (**B**) or hydHDL (**C**), the hydrolyzed lipoproteins were incubated with SAA using protein weight ratios of 1:1 SAA:apoA-I (for HDL) or 1:1 SAA:apoB (for LDL) as described in 'Materials and methods'. Similar incubation mixtures of SAA with non-hydrolyzed HDL (SAA+HDL) or LDL (SAA+LDL) are shown for comparison. *Figure 3—figure supplement 1* shows non-denaturing PAGE that monitors the remodeling of SAA-containing model and plasma lipoproteins upon their hydrolysis by sPLA$_2$.

DOI: https://doi.org/10.7554/eLife.46630.008

The following figure supplement is available for figure 3:

**Figure supplement 1.** Non-denaturing PAGE monitors remodeling of SAA-containing model and plasma lipoproteins upon their hydrolysis by sPLA$_2$.
DOI: https://doi.org/10.7554/eLife.46630.009

protein) — were hydrolyzed with sPLA$_2$ as described in *Figure 3—figure supplement 1*. The 7–7.5 nm particles formed upon hydrolysis were isolated in the density range 1.16–1.20 g/ml. In control experiments, three density fractions were taken after centrifugation of SAA-POPC complexes: before hydrolysis (1.16–1.18 g/ml), hydrolyzed SAA-POPC (1.17–1.20 g/ml), and lipid-free SAA (>1.22 g/ml). Non-denaturing PAGE detected no 7–7.5 nm species at 1.16–1.18 g/ml in the control experiments. By contrast, samples of hydrolyzed SAA-POPC clearly showed 7–7.5 nm species in the 1.17–1.20 g/ml density fraction (*Figure 4—figure supplement 1*). SAA-containing samples of hydrolyzed HDL and LDL also showed species in this range of size and density (*Figure 4—figure supplement 1B,C*). After isolation by centrifugation, the migration pattern changed slightly and smaller particles became predominant; henceforth these are collectively termed ~7 nm species.

To assess the number of SAA molecules per particle, SAA-POPC particles were cross-linked with glutaraldehyde. SDS PAGE of intact SAA-POPC showed sharp bands corresponding to protein monomers, dimers and trimers, whereas hydrolyzed SAA-POPC showed a prominent hexamer band (*Figure 4—figure supplement 1D*), suggesting that each particle contained at least six protein molecules.

We used SDS PAGE and mass spectrometry to determine the protein composition of this isolated ~7 nm species. The results showed both SAA and apoA-I in the species released from the hydrolyzed SAA-HDL (*Figure 4A,B*). Only SAA was detected in the species released from hydrolyzed LDL in the presence of SAA (*Figure 4C,D*).

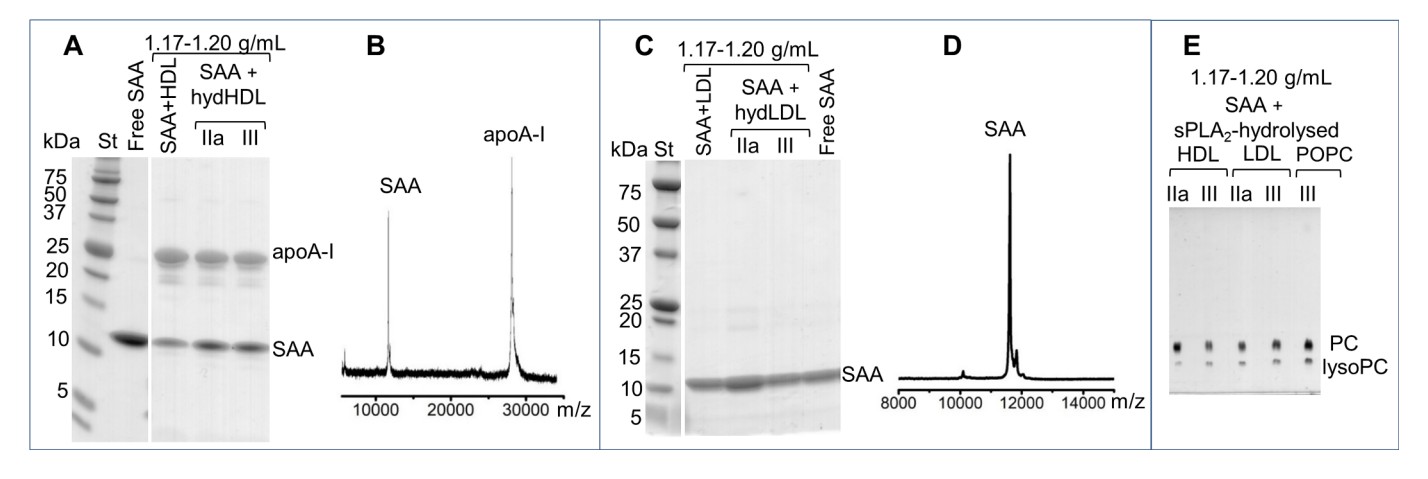

**Figure 4.** Biochemical analysis of the ~7 nm complexes formed by SAA and lipolytic products. SAA-containing complexes, which were obtained upon lipolysis of model (SAA-POPC) or plasma lipoproteins (HDL and LDL) by sPLA$_2$, were isolated at 1.17–1.20 g/mL density (*Figure 4—figure supplement 1*). These isolated complexes, marked SAA + hydHDL (A), SAA + hydLDL (B) or SAA + sPLA$_2$-hydrolyzed HDL, LDL or POPC (E), were analyzed for protein (A–D) and lipid composition (E). SDS PAGE (A) and matrix-assisted laser desorption ionization – time of flight mass spectrometry (B) of SAA + hydHDL revealed SAA and apoA-I; the protein mass detected by mass spectrometry was 11,606 Da for SAA and 28,086 for apoA-I (B). Similar analyses of SAA + hydLDL complexes showed only SAA (C, D). (E) Thin-layer chromatography showed the presence of PC and lysoPC in the SAA-containing ~7 nm complexes that were obtained from all hydrolyzed lipoproteins (HDL, LDL) or model lipids (POPC). Lipid-free SAA and the 1.17–1.20 g/mL density fraction isolated from SAA mixtures with non-hydrolyzed HDL or LDL (SAA + HDL in panel (A) and SAA + LDL in (C)) are shown for comparison.

DOI: https://doi.org/10.7554/eLife.46630.010

The following figure supplement is available for figure 4:

**Figure supplement 1.** Gel electrophoresis of isolated SAA complexes with the products of phospholipid hydrolysis.
DOI: https://doi.org/10.7554/eLife.46630.011

Lipid composition in this species was assessed by thin-layer chromatography and enzymatic assays. Both PC and lysoPC were observed in the ~7 nm species isolated from all hydrolyzed lipoproteins, including SAA-POPC, SAA-HDL, and SAA-containing LDL samples (*Figure 4E*). FFA and phospholipid assays showed 30–45% FFA and 12–22% PC as a weight fraction of total lipids in these ~7 nm species. We conclude that SAA sequesters the FFA and lysoPC that are produced upon the lipolysis of diverse lipoproteins by sPLA$_2$, and removes these hydrolytic products from the parent particle in the form of ~7 nm protein-lipid complexes. These complexes are heterogeneous and their exact size and biochemical composition vary depending on the parent lipoproteins, yet they all contain SAA and the products of lipolysis.

## SAA forms stable complexes with the products of sPLA$_2$ hydrolysis of model lipoproteins

Previous studies showed that the binding of SAA to POPC and other phospholipids induces α-helical folding in this intrinsically disordered protein at ambient temperatures, greatly increasing the thermal stability of SAA and protecting it from proteolysis (*Takase et al., 2014*; *Jayaraman et al., 2015*; *Frame et al., 2017*; *Lu et al., 2014*). To probe whether the SAA-containing complexes that are released upon hydrolysis of these precursors also formed stable structures, these ~7 nm complexes were isolated by density in the 1.17–1.20 g/ml range, and their secondary structure and stability were assessed by circular dichroism (CD) spectroscopy. Far-UV CD spectra at 25°C showed a major conformational change, from a largely unfolded secondary structure in free SAA to ~40% α-helix in complexes with lipids. Notably, the lipid-bound secondary structure was very similar in the parent SAA-POPC particles and in the ~7 nm products that are released upon lipolysis (*Figure 5A*).

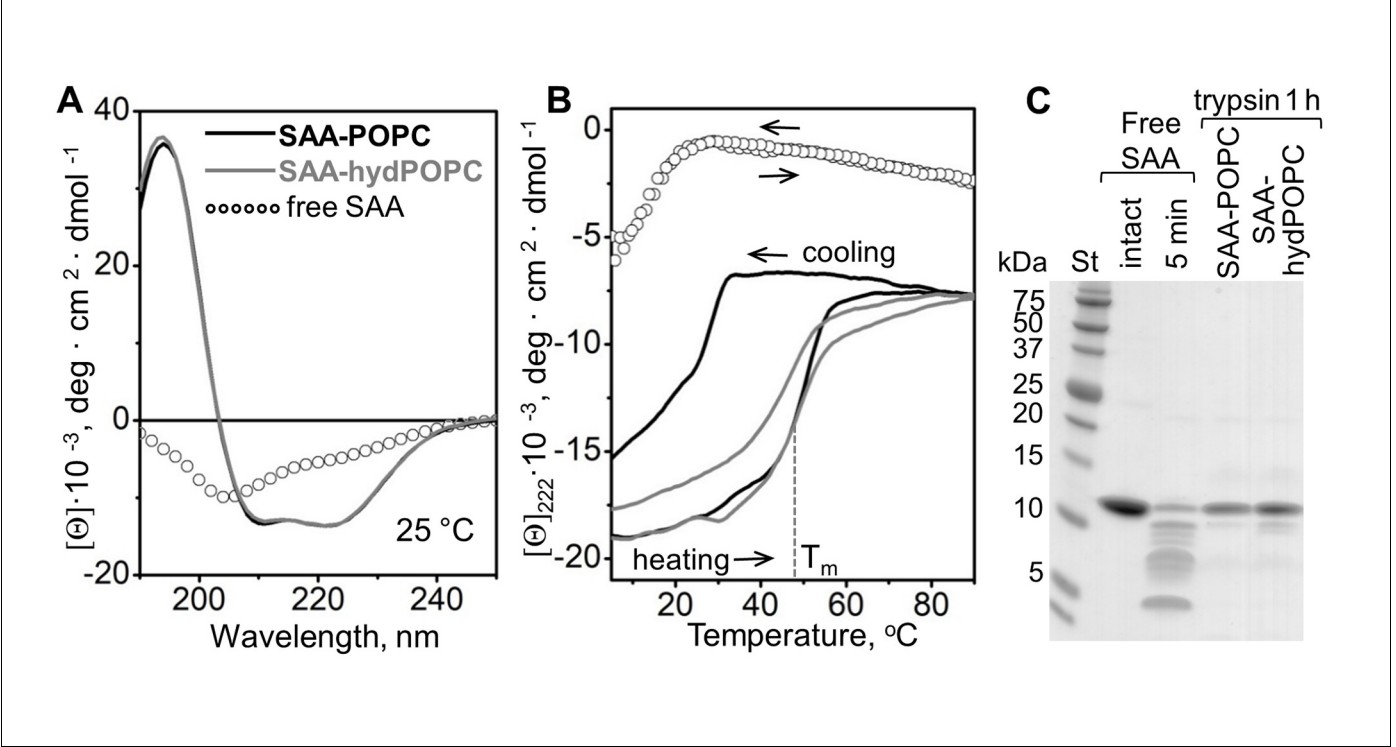

**Figure 5.** Structure and stability of SAA complexes with the products of POPC hydrolysis. Intact SAA-POPC complexes (prepared using 1:10 protein: lipid molar ratio) were hydrolyzed with sPLA$_2$-III, and the 1.17–1.20 g/ml density fraction containing ~7 nm particles was isolated (for details, see main text, *Figure 3* and 'Materials and methods'). Far-UV CD spectra at 25°C (**A**) and the melting data (**B**) are shown for these isolated ~7 nm complexes (SAA-hydPOPC), the parent particles (SAA-POPC) and lipid-free SAA. The melting data show CD signal at 222 nm, $[\Theta]_{222}$(T), that monitors α-helical structure during heating and cooling from 10°C to 90°C at a rate of 60 °C/h. Arrows show directions of the temperature changes; the dotted line indicates the melting temperature for SAA-hydPOPC, $T_m$ = 48 ± 2°C. (**C**) Limited tryptic digestion of free SAA, SAA-POPC, and SAA-hydPOPC complexes monitored by SDS PAGE. Trypsin was incubated at room temperature for 5 min with free SAA or for 1 hr with SAA-lipid complexes (see 'Materials and methods' for details). *Figure 5—figure supplement 1* shows the structure and stability of the 7–7.5 nm complexes formed by SAA and the products of LDL hydrolysis. *Figure 5—figure supplement 2* shows the structure and stability of SAA complexes with lysoPC and POPC.
DOI: https://doi.org/10.7554/eLife.46630.012

The following figure supplements are available for figure 5:

**Figure supplement 1.** Structure and stability of the 7–7.5 nm complexes formed by SAA and the products of LDL hydrolysis.
DOI: https://doi.org/10.7554/eLife.46630.013
**Figure supplement 2.** Structure and stability of SAA complexes with lysoPC and POPC.
DOI: https://doi.org/10.7554/eLife.46630.014

The structural stability of the ~7 nm products was assessed by measuring the CD signal at 222 nm as a function of temperature to monitor helical unfolding and refolding during heating and cooling from 5°C to 95°C. The heating data for the precursor particles and the ~7 nm product species partially overlapped and showed similar melting temperatures, $T_m$ = 50 ± 2°C and 48 ± 2°C, respectively (*Figure 5B*), much higher than that of free SAA ($T_m$ = 17 ± 2°C) (*Jayaraman et al., 2015*; *Frame et al., 2017*). Therefore, SAA complexes with either intact or hydrolyzed POPC showed comparable thermal stability that was much higher than that of free SAA. Unlike the heating data, the cooling data for the two complexes significantly differed (*Figure 5B*, gray and black solid lines). For the precursor SAA-POPC particles, the refolding upon cooling was observed at much lower temperatures than the unfolding upon heating. Such a hysteresis is a hallmark of thermodynamically irreversible transitions; in lipoproteins, it reflects irreversible structural remodeling such as fusion (*Jayaraman et al., 2015* and references therein). By contrast, free SAA shows a reversible unfolding without a hysteresis (*Figure 5B*, open circles). Notably, in the ~7 nm SAA complexes formed upon hydrolysis, the heating and cooling transitions were much closer than those in the precursor particles, and the hysteresis was nearly abolished (*Figure 5B*, gray solid line). This observation is

consistent with the relatively high protein to lipid ratio in the ~7 nm complexes, which is evident from their higher density and smaller size as compared to the precursor particles.

The conformational stability of the ~7 nm complexes was further probed by limited proteolysis as described in 'Materials and methods'. In contrast to free SAA, which was largely fragmented within 5 min of incubation with trypsin at 22°C (*Figure 5C*), the ~7 nm complexes and their SAA-POPC precursor particles resisted proteolysis, and they underwent no major fragmentation even after 12 hr of incubation (*Figure 5C*). These results agree with the CD data showing more helical structure and increased stability in lipid-bound SAA when compared with free SAA (*Figure 5A,B*).

Together, the results showed that SAA forms ~7 nm complexes with the hydrolytic products of sPLA$_2$ (*Figure 3—figure supplement 1*, *Figure 4—figure supplement 1*, *Figure 5*). Although these complexes migrate in the same size range as free SAA on the non-denaturing gel, their structure is distinct from that of either free SAA or the SAA-POPC precursor particles. Unlike free SAA, which shows unfolded secondary structure when examined by CD and is rapidly degraded by trypsin at ambient temperatures, SAA in the ~7 nm complexes is ~40% helical and resists proteolysis (*Figure 5A–C*), thereby resembling the SAA-POPC precursors. In contrast to precursors, these ~7 nm complexes: i) contain large amounts of FFA and lysoPC (*Figure 4*); ii) have smaller size (6.5–7.5 nm) and higher density (1.17–1.20 g/mL) indicative of a higher protein to lipid ratio; and iii) undergo a more thermodynamically reversible thermal unfolding (*Figure 5B*). The latter is consistent with the kinetically accessible character of these product complexes suggested by their similarity, regardless of the order of SAA binding and sPLA$_2$ hydrolysis (*Figure 3A*, *Figure 3—figure supplement 1A*). To our knowledge, such small stable protein-rich proteolysis-resistant complexes comprised of SAA and the products of phospholipid hydrolysis have not been reported previously.

## SAA forms stable complexes with the products of sPLA$_2$ hydrolysis of plasma lipoproteins

To ascertain that the high structural stability of the SAA complexes with the products of phospholipid hydrolysis is not limited to model systems, we analyzed the conformation and stability of the ~7 nm complexes isolated from a mixture of SAA and sPLA$_2$-hydrolyzed plasma lipoproteins. The complexes released from the hydrolyzed SAA-HDL contained both SAA and apoA-I (*Figure 4A,B*). Owing to difficulties in dissecting the contributions from individual proteins, these complexes were not explored in detail. Instead, we focused on the ~7 nm complexes released from hydrolyzed LDL in the presence of SAA. Such complexes contained SAA as their sole protein (*Figure 4C,D*), facilitating a direct comparison with a model SAA-POPC system.

Parent LDL was hydrolyzed with sPLA$_2$ and incubated with SAA, leading to the formation of 7–7.5 nm SAA-only complexes (*Figure 3C*) that were isolated by density at 1.17–1.20 g/ml. Far-UV CD spectra of these complexes revealed a ~ 40% α-helical conformation (*Figure 5—figure supplement 1A*), similar to that seen in the ~7 nm complexes released upon lipolysis of SAA-POPC (*Figure 5A*). Moreover, similar to these model complexes, thermal unfolding of the LDL-derived SAA-containing complexes was observed with T$_m$ = 45 ± 2°C and was largely thermodynamically reversible (*Figure 5—figure supplement 1B*). Finally, limited proteolysis ascertained the high conformational stability of SAA in these complexes, which resisted fragmentation upon incubation with trypsin at 22°C for up to 24 hr (*Figure 5—figure supplement 1C*).

We conclude that SAA sequesters the hydrolytic products of sPLA$_2$ from model and plasma lipoproteins and forms stable ~7 nm complexes with these products. These complexes migrate in the same size range on the non-denaturing gel as free self-associated mSAA1.1. However, unlike free SAA that is structurally labile, the protein in these complexes acquires a stable highly α-helical proteolysis-resistant conformation at ambient temperatures.

## SAA forms stable binary complexes with FFA and with lysoPC

Hydrolysis of POPC by sPLA$_2$ generates equimolar amounts of oleic acid (OA) and lysoPC. Upon incubation with OA, SAA was previously shown to form spontaneously binary SAA-OA complexes that migrate at ~7.5 nm; the protein in these complexes was ~40% α-helical at 25°C and resisted tryptic digestion (*Jayaraman et al., 2018*). Here, we explored the formation and properties of binary complexes of SAA with lysoPC. The complexes were formed as described in the 'Materials and methods' by incubating SAA with lysoPC at 1:10 protein to lipid molar ratio; similar experiments

with POPC were used as a control. Non-denaturing PAGE showed SAA-lysoPC particles of ~8 nm in size, similar to those of SAA-POPC formed at 1:10 molar ratio of protein to lipid (*Figure 5—figure supplement 2A*). Far-UV CD spectra showed substantial α-helical content in SAA-lysoPC particles, slightly lower than that in SAA-POPC particles (35% versus 40%) (*Figure 5—figure supplement 2B*). Thermal unfolding of SAA-lysoPC particles was observed with $T_m$ = 45°C, slightly lower than 50°C seen in SAA-POPC (*Figure 5—figure supplement 2C*). Unlike SAA-POPC particles, SAA-lysoPC particles showed little hysteresis during thermal unfolding and refolding. Moreover, similar to other SAA-lipid complexes, SAA-lysoPC complexes resisted tryptic digestion (*Figure 5—figure supplement 2D*).

Together, our results showed that SAA forms binary complexes with OA and with lysoPC. The protein in these complexes is 35–40% α-helical and resists proteolysis at ambient temperatures; the thermal unfolding is centered at $T_m$ = 45–48°C and is largely thermodynamically reversible. In this regard, these binary complexes of SAA-OA and SAA-lysoPC resemble the quaternary complexes containing SAA, OA, lysoPC and POPC (*Figure 5*). Clearly, SAA can sequester POPC and its hydrolytic products, either separately or together, to form highly α-helical proteolysis-resistant complexes that are thermodynamically stable at ambient temperatures.

## Relevance to disease and comparison with FFA removal by albumin

Next, we tested whether SAA can sequester the naturally occurring hydrolytic products from human lipoproteins. Our focus was on the FFA that are elevated in diabetes, inflammation and other diseases. LDL was isolated from the pooled plasma of subjects who were either normolipidemic or had diabetes mellitus, as previously described (*Jayaraman et al., 2017b*). The content of endogenous FFA in these diabetic LDL was 20% higher than that in normolipidemic LDL (*Jayaraman et al., 2017b*). SAA was incubated with LDL using 1:1 weight ratio of SAA to apoB. Non-denaturing PAGE showed that the SAA-containing ~7.5 nm complexes were released from diabetic as well as normolipidemic LDL (*Figure 6—figure supplement 1A*), similar to those released from LDL upon hydrolysis by sPLA2 in the presence of SAA (*Figure 3C, Figure 2—figure supplement 1D*). This result suggests that SAA sequesters exogenous and endogenous FAA from lipoproteins to form similar-size complexes.

Under normal in vivo conditions, FFA and lysoPC are transported in plasma mainly by serum albumin. In acute inflammation, plasma levels of albumin and its ability to sequester FFA and lysoPC decrease, while the levels of sPLA2 and SAA increase (*Gabay and Kushner, 1999*; *Pruzanski and Vadas, 1991*; *Fichtlscherer et al., 2004*). Is it possible for SAA to assume albumin's function under these conditions? To explore this, we compared the ability of SAA and albumin to remove endogenous FFA from plasma LDL. LDL from diabetic and healthy normolipidemic subjects (0.2 mg/ml apoB) was incubated at pH 7.5, 37°C, for 6 hr with 0.2 mg/ml SAA (2% w/v). A similar LDL incubation was carried out with 2% w/v human serum albumin. As a control, LDL was incubated under similar conditions without any added proteins. LDL was isolated by SEC and the FFA content was determined. The results showed that both albumin and SAA removed a large portion of endogenous FFA from diabetic as well as normolipidemic LDL, and that albumin was slightly more efficient than SAA at pH 7.5 (*Figure 6—figure supplement 1B*). This finding suggests that SAA can potentially contribute to FFA removal at plasma pH, albeit less efficiently than albumin.

Next, we tested whether SAA can remove FFA at acidic pH, which severely impairs albumin's function (*Lähdesmäki et al., 2009*). Previously, we showed that the SAA structure, stability and ability to remodel POPC MLV remain invariant at pH 7.5–5.5 but are altered at near-lysosomal pH (*Jayaraman et al., 2017a*). Here, we explored how pH influences the ability of SAA to remove FFA from human lipoproteins. Single-donor LDLs were hydrolyzed with sPLA2 and incubated either with SAA, with albumin, or alone (control) for 6 hr at pH ranging from 4.5 to 7.5. Thereafter, LDL was isolated by density gradient centrifugation and the FFA content was measured. As expected, albumin was less efficient at removing FFA at acidic pH than at pH 7.5; this was evident from the higher residual content of FFA detected in LDL at pH 4.5–6.5 versus that at pH 7.5 (*Figure 6*, LDL + albumin). By contrast, SAA's ability to remove FFA remained invariant at pH 4.5–7.5 (*Figure 6*, LDL + SAA). Importantly, at acidic pH, when the activity of albumin but not SAA was impaired, SAA removed more FFA than albumin per gram of protein (*Figure 6*, LDL + SAA and LDL + albumin compared). This finding strongly supports the role of SAA in FFA removal in vivo and suggests that this role becomes particularly important at the acidic pH that are typical ofinflammation sites. This role is

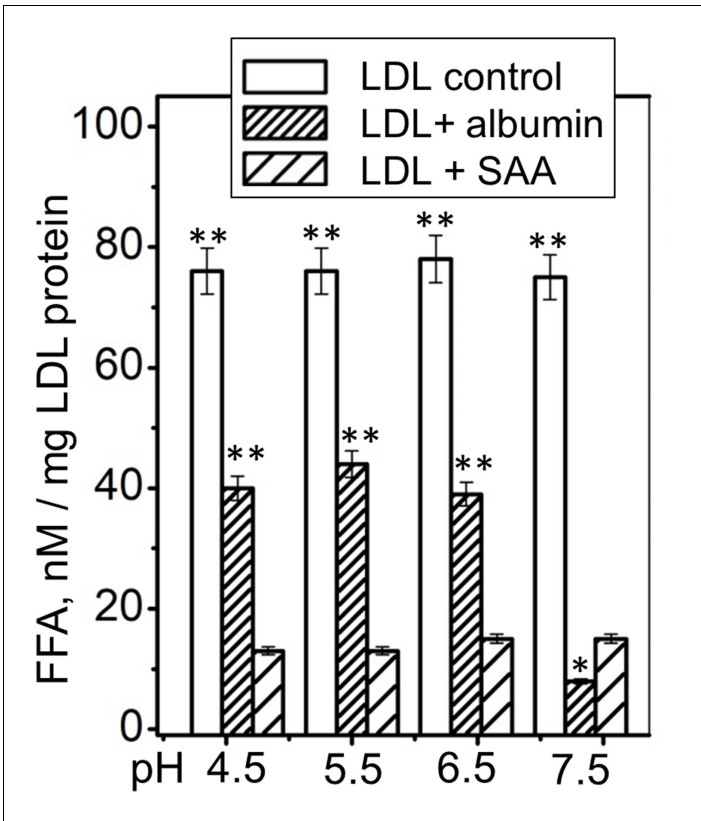

**Figure 6.** FFA removal by SAA and by albumin as a function of pH. Single-donor LDL were hydrolyzed with sPLA$_2$-III and incubated at pH 4.5–7.5 either alone (control), with albumin (LDL + albumin) or with SAA (LDL + SAA). Next, LDL were re-isolated by density gradient centrifugation and their residual FFA content was determined. The results are shown as the mean of three independent measurements with their standard errors. FFA levels in controls and in albumin-treated LDL were compared with those in SAA-treated LDL using one-way ANOVA; *, p<0.5; **, p<0.01. *Figure 6—figure supplement 1* shows that SAA and albumin sequester naturally occurring FFA from human plasma LDL at pH 7.5. *Figure 6—figure supplement 2* shows that human SAA 1.1 sequesters the products of phospholipid hydrolysis from model and plasma lipoproteins.
DOI: https://doi.org/10.7554/eLife.46630.015

The following source data and figure supplements are available for figure 6:

**Source data 1.** Free fatty acid analysis of LDL at different pH.
DOI: https://doi.org/10.7554/eLife.46630.020

**Figure supplement 1.** SAA and albumin sequester naturally occurring FFA from human plasma LDL at pH 7.5.
DOI: https://doi.org/10.7554/eLife.46630.016

**Figure supplement 1—source data 1.** Free fatty analysis of normal and diabetic LDL.
DOI: https://doi.org/10.7554/eLife.46630.017

**Figure supplement 2.** Non-denaturing PAGE shows that human SAA 1.1 (hSAA1.1) sequesters the products of phospholipid hydrolysis from model and plasma lipoproteins.
DOI: https://doi.org/10.7554/eLife.46630.018

**Figure supplement 2—source data 1.** Free fatty acid analysis of LDL incubated with human and mouse SAA.
DOI: https://doi.org/10.7554/eLife.46630.019

expected to be important in inflammation and in other conditions in which SAA level increases while albumin level and activity declines.

## Human and murine SAA1.1 sequester the products of phospholipid hydrolysis

To ascertain the relevance of these findings to humans, selected experiments were performed using hSAA1.1 and either model or plasma lipoproteins that were hydrolyzed by sPLA$_2$. Human and

murine SAA1.1 show striking sequence, structural and functional similarities (*Rennegarbe et al., 2017*). Compared to mSAA1.1, hSAA1.1 is less water-soluble in the lipid-free state and forms larger ~9 nm oligomers on the non-denaturing PAGE (hSAA, *Figure 6—figure supplement 2A*) relative to the ~7.5 nm oligomers formed by free mSAA1.1 (*Figure 1B*, *Figure 4—figure supplement 1A*, *Figure 6—figure supplement 1A*). Importantly, non-denaturing PAGE showed that, similar to mSAA1.1, hSAA1.1 spontaneously formed 8.5–9 nm particles upon incubation with POPC SUV at a 1:10 protein to lipid molar ratio (hSAA-POPC, *Figure 6—figure supplement 2A*). Upon hydrolysis with sPLA$_2$-IIa or sPLA$_2$-III, these 8.5–9 nm particles were remodeled into ~7.5 nm complexes (sPLA$_2$-hydrolyzed, *Figure 6—figure supplement 2A*), similar to those formed by mSAA1.1 (sPLA$_2$-hydrolyzed 1:10, *Figure 3—figure supplement 1A*).

Further, when hSAA1.1 was incubated with LDL that have been hydrolyzed with sPLA$_2$, it also formed complexes that migrated at ~7.5 nm (hSAA + LDL, *Figure 6—figure supplement 2B*). Complexes of similar size were formed by mSAA1.1 upon sequestration of lipolytic products from LDL (SAA + hydLDL, *Figure 4—figure supplement 1C*). These comparisons suggest that, similar to mSAA1.1, hSAA1.1 sequesters the products of phospholipid hydrolysis from model and plasma lipoproteins to form ~7.5 nm particles. Finally, FFA analysis of LDL from the plasma of normolipidemic and diabetic patients that have been incubated with SAA and then re-isolated by density showed that, similar to mSAA1, hSAA1.1 sequestered a major fraction of FFA from these LDL (*Figure 6—figure supplement 2C*). Together, these results suggest that, similar to mSAA1.1, hSAA1.1 can sequester phospholipids and their lipolytic products into HDL-size particles.

## Discussion

This in vitro study demonstrates that two ancient acute-phase plasma proteins, sPLA$_2$ and SAA, act in synergy to break down and remove phospholipids. The increased lipolytic activity of sPLA$_2$ on HDL upon the addition of SAA was previously reported but the mechanism was unknown (*Pruzanski et al., 1995*). The current study shows that SAA enhances the lipolysis by sPLA$_2$ of diverse lipid assemblies, including model phospholipid vesicles as well as human HDL and LDL (*Figures 1* and *2*), and that the mechanism involves direct SAA-mediated enhancement of sPLA$_2$ activity. We show that SAA augments the sPLA$_2$ reaction both by generating highly curved substrates for sPLA$_2$ (*Figure 1*) and by removing its reaction products, FFA and lysophospholipids (*Figure 4*). The latter may well be particularly important because product removal determines the reaction rate of many lipases including sPLA$_2$ (*Carman et al., 1995*). Together, these findings expand the previously postulated housekeeping roles of SAA in solubilizing and clearing cellular membrane phospholipids from injured sites (*Frame et al., 2017*) and in providing an anti-oxidant for lipoproteins by sequestering lipid peroxides (*Jayaraman et al., 2016*). Our results suggest that sequestration of membrane lipids and their degradation products by SAA and the safe removal of these bioactive products from the injured sites, which is a prerequisite for tissue healing, represents a vital role of this Cambrian protein.

Unlike SAA, several other acute–phase reactants either have no effect on sPLA$_2$ activity or inhibit it, which perhaps helps to control the resolution of inflammation (*Pruzanski et al., 1996*). To our knowledge, SAA is the only acute-phase protein that enhances the activity of sPLA$_2$. The in vivo synergy between SAA and sPLA$_2$ is probably facilitated by the simultaneous secretion of these proteins systemically as well as locally at the inflammation sites. Furthermore, both proteins show preference for highly curved lipid surfaces that can be generated by SAA (*Figure 1*; *Frame et al., 2017*; *Jayaraman et al., 2018*) and are required for the activation of sPLA$_2$. This is expected to lead to spatiotemporal overlap between SAA and sPLA$_2$, facilitating their synergy at the inflammation sites.

We propose that the ability of SAA to sequester diverse phospholipids and their degradation products, which was demonstrated in the current and previous studies by us and others (*Takase et al., 2014*; *Tanaka et al., 2017*; *Frame et al., 2017*; *Jayaraman et al., 2018*; *Jayaraman et al., 2016*), underlies several beneficial effects. First, SAA can spontaneously solubilize diverse phospholipid bilayers in vitro (*Figure 1A,B*; *Jayaraman et al., 2018*), and perhaps in vivo, which is particularly relevant with respect to dead cells whose normal lipid efflux to HDL via ATP-driven transporters is impaired (*Frame et al., 2017*). By contrast, lipid sequestration by SAA is energy-independent (*Figure 1A*). The resulting stable nanoparticles are hydrolyzed by sPLA$_2$ (*Figure 1C*; also see *Jayaraman et al., 2018*) and/or perhaps are internalized by

macrophages through scavenger receptors such as CD36 or SR-BI that bind SAA (*Eklund et al., 2012*; *Frame et al., 2017* and references therein). Second, SAA solubilizes lysoPC and FFA and sequesters them into small particles (*Figure 3*) that are stable, substantially α-helical, and resist proteolysis at 37°C (*Figure 5*). This sequestration augments the sPLA$_2$ reaction; it is also expected to facilitate the safe removal of toxic lipolytic products by SAA while protecting free SAA from unfolding and rapid degradation. Third, consistent with these findings, previous studies showed that SAA protects human lipoproteins from oxidation in vitro (*Jayaraman et al., 2016*) and in vivo (*Sato et al., 2016*). This anti-oxidant effect is mediated mainly by free (rather than HDL-bound) SAA that probably sequesters lipid hydroperoxides and safely removes them from lipoproteins (*Jayaraman et al., 2016*). in a manner similar to the SAA-mediated removal of FFA and lysoPC described in the current study. Together, these findings suggest that during inflammation, when oxidative stress and lipolysis are increased, SAA augments the action of serum albumin by removing FFA, lysoPC, lipid peroxides and, potentially, other products of lipid degradation. We speculate that free SAA, like albumin, serves as a lipid scavenger that sequesters diverse lipophilic molecules. Unlike albumin, SAA forms oligomers to sequester lipids. Although the structure of these oligomers is unknown, lipids are expected to bind in a hydrophobic cavity formed by concave apolar faces of amphipathic helices from several protein molecules (*Frame and Gursky, 2016*; *Frame et al., 2017*). The high stability of SAA complexes with lipids and their hydrolytic products contrasts with the marginal in vitro stability of lipid-free SAA oligomers at pH above pH 5 (*Jayaraman et al., 2017a* and references therein). We speculate that these marginally stabile protein oligomers are primed for interaction with lipids to form stable complexes like those reported in the current study.

Normally, serum albumin removes a major fraction of sPLA$_2$ products. However, albumin is a negative acute-phase reactant whose plasma levels drop sharply in inflammation, with a concomitant steep increase in positive reactants, such as SAA and sPLA$_2$ (*Gabay and Kushner, 1999*). In addition, albumin's ability to remove FFA is compromised upon the protein's post-translational modifications in inflammation (*Lee and Wu, 2015*). Moreover, albumin's capacity to remove FFA decreases under acidic conditions (*Figure 6*), which are characteristic of inflammation sites (*Lähdesmäki et al., 2009*). All together, these effects are expected to cause an imbalance between the massive generation of FFA and lysoPC at inflammation sites and the impaired removal of these molecules by albumin. Our finding that, per gram of protein, SAA is more efficient than albumin in removing FFA from plasma lipoproteins at acidic pH (*Figure 6*) suggests that SAA acts locally at the sites of inflammation to compensate for the impaired albumin activity, and helps to remove excess lipids and the products of their hydrolysis and oxidation. We propose that the ability to sequester diverse lipids and their degradation products, which is rooted in the unique structure of SAA with its concave hydrophobic surface that has been highly conserved throughout evolution (*Frame and Gursky, 2016*), constitutes a previously unknown vital role of SAA in the immune response. This role remains to be tested in future cell-based and in vivo studies.

## Materials and methods

### Proteins and lipids

Recombinant murine SAA isoform 1.1 (103 amino acids, 11.6 kDa) was used throughout this study; it is termed mSAA1.1, or SAA for brevity. SAA was expressed in *Escherichia coli* and purified to 95% purity as described previously (*Kollmer et al., 2016*). In selected experiments, we used recombinant human SAA isoform 1.1 (hSAA1.1, 104 amino acids, cat. # SRP4324) from Sigma. Essentially fatty acid-free human serum albumin (cat. # A1887) was from Sigma. Lipids 1-palmitoyl-2-oleoyl-*sn*-glycero-3-phosphocholine (POPC; C16:0, C18:1) and lysoPC (16:0) were 97% + pure from Avanti Polar Lipids. Trypsin from bovine pancreas (cat. # T1426), group-III sPLA$_2$ (sPLA$_2$-III, cat. # P9279), lipoprotein lipase (cat. # L2254), sphingomyelinase from *Bacillus cerus* (cat. # S9396), and sodium oleate (cat # O7501) were from Sigma. Human recombinant group-IIa sPLA$_2$ (sPLA$_2$-IIa, cat. # RD172054100) was from Biovendor. Enzychrom free fatty acid assay kit (cat. # EFFA-100) and Enzychrom phospholipid assay kit (cat. # EPLP-100) were from Fisher Scientific. Ultrapure sodium phosphate buffer at pH 7.5 (BB-148) was from Boston Bioproducts. All other chemicals were of the highest purity analytical grade.

SAA stock solutions were prepared daily by dissolving lyophilized protein at 1 mg/ml in water and dialyzing it overnight against the standard buffer (50 mM sodium phosphate, 150 mM NaCl, pH 7.5). SAA stock solutions were centrifuged at 10,000 g for 10 min prior to each experiment to remove protein aggregates. Recombinant hSAA1 was reconstituted according to the manufacturer's recommendations and was immediately diluted in standard buffer, followed by overnight dialysis in standard buffer prior to use. Protein concentrations were determined by a bicinchoninic acid assay.

## Human plasma lipoproteins

Unless otherwise stated, single-donor human lipoproteins from three healthy volunteers were used throughout this study. Plasma from anonymous healthy donors was obtained commercially from the local blood bank according to the rules of the institutional review board. Single-donor lipoproteins were isolated following published protocols (*Schumaker and Puppione, 1986*) from fresh EDTA-treated plasma by KBr density gradient ultracentrifugation in the density range 0.94–1.006 g/mL for VLDL, 1.019–1.063 g/mL for LDL, and 1.063–1.21 g/mL for HDL. Lipoproteins from each class migrated as a single band on the agarose and non-denaturing gels. Lipoprotein stock solutions were prepared by extensive dialysis against 50 mM sodium phosphate buffer, 150 mM NaCl, 0.25 mM EDTA, 0.02% NaN$_3$, pH 7.5, degassed, and stored in the dark at 4℃. Each stock solution was used within two weeks during which no protein degradation was detected by SDS PAGE and no changes in the lipoprotein electrophoretic mobility were observed by agarose PAGE.

To obtain HDL enriched with exogenous SAA (termed SAA-HDL), single-donor HDL were isolated from healthy subjects and were incubated with SAA at 37℃ for 6 hr in standard buffer. The molar ratio of SAA to apoA-I varied from 0:1 (SAA-free control) to 4:1. Free (uncomplexed) proteins, which contained excess SAA as well as apoA-I that was displaced from HDL by SAA, were removed by size exclusion chromatography (SEC) as previously described (*Jayaraman et al., 2015*). The total incubation mixture before purification is termed SAA-HDL(total) and the SEC-purified lipoprotein fraction is termed SEC Fr (*Figure 2—figure supplement 1A,B*). As a control, HDL were incubated without SAA under otherwise identical conditions; no changes in the particle size, composition or stability were detected upon such incubation. Total protein concentration was determined by bicinchoninic acid assay, and individual apolipoprotein content was determined as a weight fraction of the total protein by quantifying SDS gel bands using image J software, as described previously (*Jayaraman et al., 2015*). Intact non-modified HDL contained 75% apoA-I and 25% apoA-II; SAA-HDL (1:1 mol:mol SAA:apoA-I, SEC fr) contained 65% apoA-I, 23% apoA-II and 12% SAA; and SAA-HDL (4:1, SEC fr) contained 56% apoA-I, 17% apoA-II and 27% SAA (*Figure 2—figure supplement 1C*).

To explore the effects of SAA on LDL, SAA (1 mg/ml) was incubated with single-donor nomolipidemic plasma LDL (1 mg/ml apoB) in standard buffer at 37℃ for 3 hr. The mixture was analyzed by SEC and non-denaturing PAGE to ascertain that SAA did not bind LDL (*Figure 2—figure supplement 1D*).

To compare human LDLs that vary in endogenous FFA levels, plasma was pooled from 25 patients diagnosed with type-2 diabetes mellitus and from 25 healthy normolipidemic subjects, as previously described (*Jayaraman et al., 2017b*). Plasma was obtained at the Lipid Laboratory of Hospital de Saint Pau and LDL were isolated by density gradient centrifugation in the laboratory of Dr. Jose Luís Sanchez-Quesada at the Hospital de Saint Pau (Barcelona, Spain); these studies were done with the written informed consent of the patients and upon approval by the institutional ethics committee, as previously described (*Jayaraman et al., 2017b*). The FFA content, which was quantified by an enzymatic assay described below, was 20% higher in diabetic than in normolipidemic LDL.

## Reconstituted lipoproteins

Lipoproteins were reconstituted using a thin film evaporation method. POPC was dissolved in chloroform:methanol (2:1 v/v), the organic solvent was evaporated under nitrogen stream, and the samples were dried under vacuum overnight at 4℃. MLVs were prepared by dispersing the lipid film in standard buffer followed by vortexing. SUVs (diameter ~22 nm) were prepared by sonication of MLVs and were used within one day.

SAA-containing lipoproteins of controled size were prepared by incubating SAA with POPC SUV at 25℃ for 3 hr using an SAA to POPC molar ratio of either 1:100 (to prepare SAA-containing SUVs of ~22 nm in size) or 1:10 (to prepare SAA-POPC complexes of 8 nm in size). Excess lipid was

removed by centrifugation and excess protein was removed by SEC. The protein to lipid weight ratio in the final preparations was 1:42 (for 1:100 mol:mol initial ratio) or 1:28 (for 1:10 mol:mol initial ratio).

SAA complexes with oleic acid were reconstituted as described by *Jayaraman et al. (2018)*. To reconstitute SAA complexes with lysoPC, SAA was incubated with freshly prepared lysoPC at 37°C for 1 hr at protein:lipid molar ratios varying from 1:1 to 1:100. This range encompassed lysoPC concentrations below and above its critical micelle concentration (4–8.3 mM).

## Lipid clearance

MLV clearance by SAA at room temperature was monitored by turbidity at 325 nm using a Varian Cary-300 UV-vis spectrophotometer. SAA (20 µg/ml) was rapidly mixed with MLV suspension in standard buffer (40 µg/ml lipid), and the time course of turbidity changes was recorded as micron-size MLVs were converted into smaller lipoprotein nanoparticles. MLVs alone were used as controls.

## Enzymatic lipolysis

SAA-POPC complexes (10 µM lipid), as well as human plasma HDL and LDL (0.5–1.0 mg/ml total protein), were used as substrates for sPLA$_2$. First, the lipoproteins were dialyzed against 10 mM Tris buffer saline at pH 7.5. Phospholipid lipolysis was performed using 50 nm human recombinant sPLA$_2$-IIa or bee-venom sPLA$_2$-III in TBS at pH 7.4 in the presence or absence of 2% w/v of fatty-acid-free bovine serum albumin and 2 mM CaCl$_2$. After a 3 hr incubation at 37°C, the reaction was terminated by adding EDTA (final concentration 20 mM). To assess the spontaneous lipolysis of phospholipids, control experiments were carried out under identical conditions without sPLA$_2$. Hydrolysis of POPC SUV by sPLA$_2$ without SAA was also quantified as a control. The extent of hydrolysis was assessed by measuring the FFA products using an enzymatic assay kit (Wako Chemicals).

## Lipid assays

FFA and total phospholipids were quantified using kits from Bioassay Systems (EnzycrhomTM free fatty assay kit EFFA-100, and EnzychromTM phospholipid assay kit EPLP-100), according to the manufacturer's instructions.

Lipoprotein lipids were assayed by thin layer chromatography using samples containing 0.5 mg/ml protein. The lipids were extracted following published protocols (*Folch et al., 1957*), the organic solvent was dried, and the lipids were spotted on the plate (glass backed, plain silica gel). The tank was first saturated with a chloroform:methanol:water (32.5:12.5:2 v/v/v) solvent system. The plate was developed for 1 hr. The spots were identified by charring with sulfuric acid spray.

## Circular dichroism spectroscopy

CD data were recorded using an AVIV 62DS spectropolarimeter to monitor protein secondary structure and thermal stability. Far-UV CD spectra were recorded at 190–250 nm from solutions containing 0.1 mg/ml SAA in standard buffer. Melting data were recorded at 222 nm to monitor changes in the α-helical structure during sample heating and cooling at a constant rate of either 70 °C/h or 10 °C/h. The data at these scan rates closely overlapped, consistent with previous studies (*Jayaraman et al., 2015*). Buffer baselines were subtracted from the data; the results were normalized to the protein concentration and reported as molar residue ellipticity, [Θ]. Helical content was estimated on the basis of the value of [Θ]$_{222}$, as previously described (*Mao and Wallace, 1984*).

## Gel electrophoresis and size-exclusion chromatography

For non-denaturing PAGE, Novex 4–20% Tris-glycine gels (Invitrogen) were loaded with 6 µg protein per lane and run to termination at 1500 V·h under non-denaturing conditions in Tris-glycine buffer. For SDS PAGE, Novex 16% or 18% Tris-glycine gels were loaded with 5 µg protein per lane and run at 200 V for 1 hr under denaturing conditions in SDS-Tris-glycine buffer. The gels were stained with Denville Blue protein stain (Denville Scientific).

SEC was performed with a Superose 6 10/300 GL column controlled by an ÄKTA UPC 10 FPLC system (GE Healthcare). Elution by 10 mM PBS at pH 7.5 was carried out at a flow rate of 0.5 ml/min.

## Protein cross-linking

SAA oligomerization in SAA-POPC complexes before and after their hydrolysis with sPLA$_2$ was assessed using glutaraldehyde, which can cross-link free NH$_2$ groups that are separated by up to 12 Å. SAA-POPC (0.5 mg/ml protein) was incubated for 30 min at 24°C with glutaraldehyde (0.01–0.08%), the reaction was quenched by adding 100 mM Tris, and the samples were analyzed by SDS PAGE. The results were obtained using standard buffer (50 mM sodium phosphate buffer, 150 mM NaCl, pH 7.5), which yielded less non-specific aggregation. The exact cross-linking pattern varied depending on the salt concentration, and more non-specific higher order oligomers were observed in a low-salt buffer.

## Limited proteolysis and mass spectrometry

SAA, either in lipid-free form or in complexes with lipids, was incubated with trypsin at 1:1500 mg:mg enzyme:substrate ratio in standard buffer at room temperature. Tryptic digestion was quenched using 2 mM of a serine protease inhibitor, phenylmethylsulfonyl fluoride. The reaction was quenched after 5 min for lipid-free SAA, which was rapidly digested, and after 1 hr for lipid-bound SAA, which was digested much more slowly. The reaction products were analyzed by SDS PAGE and matrix-assisted laser desorption ionization – time of flight mass spectrometry.

For mass spectrometry, the spectra were recorded on a Reflex-IV spectrometer (Bruker Daltonics, Billerica, MA) equipped with a 337 nm nitrogen laser. The instrument was operated in the positive-ion reflection mode at 20 kV accelerating voltage with time-lag focusing enabled. Calibration was performed in linear mode using a standard calibration mixture containing the oxidized B-chain of bovine insulin, equine cytochrome C, equine apomyoglobin, and bovine serum albumin. The matrix, cyano-4-hydroxycinnamic acid (alpha cyano, Mw = 189 g/mol), was prepared as a saturated solution in 70% acetonitrile and 0.1% trifluoroacetic acid in water. Mass spectrometry results were reported as an average of three independent experiments.

## Removal of free fatty acids

To compare the ability of SAA and albumin to remove FFA from LDL in a broad range of physiologically relevant pH, samples of single-donor human LDL containing 1 mg/ml apoB were hydrolyzed using 50 nM sPLA$_2$-III at 37°C for 3 hr at pH 7.5, 6.5, 5.5 or 4.5. The hydrolysis was performed either sPLA$_2$-III alone, sPLA$_2$-III with 2% (w/v) FFA-free human serum albumin, or sPLA$_2$-III with 2% (w/v) SAA. After incubation, LDL were re-isolated by density gradient centrifugation, and their FFA content was quantified by an enzymatic assay. The FFA content in intact LDL from the same batch was also quantified and the results are shown in *Figure 6*.

To compare the ability of human and murine SAA1.1 to remove endogenous FFA from LDL, pooled plasma LDL from healthy normolipidemic subjects and from diabetic patients were used. LDL were incubated at 37°C for 3 hr at pH 7.5 with either hSAA1.1 or mSAA1.1 using a 1:1 SAA:apoB weight ratio. Thereafter, LDL were re-isolated by density centrifugation and their FFA content was measured. The FFA content of intact LDL from the same batch was also measured (*Figure 6—figure supplement 2C*).

## Reproducibility

To ensure reproducibility, all experiments in this study were performed three or more times, unless otherwise stated. The enzymatic assays were performed in technical triplicates of biological duplicates and are reported as an average of a technical triplicate with the corresponding standard errors of means. Statistical analysis was performed using the ANOVA t-test.

## Acknowledgements

We thank Dr. R Andrew Zoeller (Boston University School of Medicine, Boston USA) for help with thin layer chromatography, and Dr. Jose Luís Sanchez-Quesada (Hospital de Saint Pau, Barcelona, Spain) for providing pooled LDL from diabetic and normolipidemic subjects. We are indebted to Dr. Barbara M Schreiber (Boston University School of Medicine, Boston USA) for useful comments on the manuscript prior to submission. This study was supported by the National Institutes of Health

grant GM067260, by the Stewart Amyloidosis Endowment Fund, and by Deutsche Forschungsge-meinschaft grant FA456/15-1.

## Additional information

### Funding

| Funder | Grant reference number | Author |
|---|---|---|
| National Institutes of Health | GM067260 | Shobini Jayaraman<br>Olga Gursky |
| Deutsche Forschungsge-meinschaft | FA456/15-1 | Marcus Fändrich |
| Stewart Amyloidosis Endow-ment Fund | | Olga Gursky |

The funders provide resource to study design, data collection and submit the work for publication.

### Author contributions
Shobini Jayaraman, Conceptualization, Software, Formal analysis, Validation, Visualization, Methodology, Writing—original draft, Project administration, Writing—review and editing; Marcus Fändrich, Resources, Funding acquisition; Olga Gursky, Resources, Formal analysis, Supervision, Funding acquisition, Investigation, Writing—original draft, Project administration, Writing—review and editing

### Author ORCIDs
Shobini Jayaraman (iD) https://orcid.org/0000-0003-1616-5347
Olga Gursky (iD) https://orcid.org/0000-0002-8598-4824

### Decision letter and Author response
Decision letter https://doi.org/10.7554/eLife.46630.023
Author response https://doi.org/10.7554/eLife.46630.024

## Additional files

### Supplementary files
• Transparent reporting form
DOI: https://doi.org/10.7554/eLife.46630.021

### Data availability
All data generated or analysed during this study are included in the manuscript and supporting files.

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
