## [Decision Letter]

Thank you for submitting your article "Synergy between serum amyloid A and secretory phospholipase A_2_ suggests a key role of ancient protein in lipid clearance" for consideration by *eLife*. Your article has been reviewed by three peer reviewers, and the evaluation has been overseen by a Reviewing Editor and Vivek Malhotra as the Senior Editor. The following individuals involved in review of your submission have agreed to reveal their identity: Matthew A Mitsche (Reviewer #1); Vladimir N Uversky (Reviewer #2); Wilfredo Colon (Reviewer #3).

The reviewers have discussed the reviews with one another and the Reviewing Editor has drafted this decision to help you prepare a revised submission.

Summary:

This interesting study reports on how serum amyloid A (SAA) and secretory phospholipase A_2_ (sPLA_2_) act in concert to clear cell membrane debris from injury sites. The authors show that SAA solubilizes lipid bilayers and converts them into lipoprotein particles that are then acted on and hydrolyzed by sPLA_2_. All three reviewers thought that the experiments were well-executed and that the results suggested a compelling model for lipid clearance at sites of inflammation. They agreed that the findings were significant, but also raised some concerns about the purely in vitro nature of this study. They welcomed a revised submission where the following points have been addressed.

Essential revisions:

1) A major component of the author's model is the pH dependence of this process. However, this point is not rigorously developed (Figure 6 shows that SAA can remove FFA at acidic pH, but it does not show and pH dependence of SAA). Further examination of the pH dependence of SAA dependent remodeling of MLV would strengthen their hypothesis.

2) Are the 7 nm particles micelles or lamellar structures? EM structural analysis would be very informative. It is not clear why the 7 nm particle would be a better vehicle for retrieving cell debris than either an emulsion like LDL/vLDL/HDL or lamellar HDL. Addressing this by experiments or in the Discussion would increase the impact of their model.

3) Based on the large particle size, what would be the oligomeric state of mouse and human SAA in these structures, how does it compare to the oligomeric structures seen for mouse and human SAA in vitro? Are these potentially the oligomeric structures alluded to in the following statement: "Unlike albumin, SAA forms oligomers to sequester lipids. Although the structure of these oligomers is unknown, lipids are expected to bind in a hydrophobic cavity formed by concave apolar faces of amphipathic helices from several protein molecules (Frame and Gursky, 2016; Frame et al., 2017)."

4) The authors state that MLVs are not hydrolyzed by sPLA_2_ in the absence of SAA. Data should be provided to support this statement.

5) The authors state that PLA_2_ incubation with SUVs leads to hydrolysis of 40-50% of POPC, how does this compare to SAA-treated MLVs in terms of percentage. In other words, in Figure 1C, what percentage of total oleic acid does that represent?

6) Between the first and second section of the Results the authors switch from MLVs to SUVs. This should be discussed or justified. Otherwise it is difficult to relate the first portion of the manuscript to the subsequent analysis.

7) The studies of isolated HDL and LDL would be clarified by comparison to model membranes that more closely resemble the particles, for example ApoA-1 mimetic nano-particles and emulsion particles.

8) The blots in Figure 3 are difficult to see. Please provide a longer exposure, or increased contrast?

9) The formation of hyperstable complexes between SAA and lipid hydrolysis products is opposite to the known marginal stability of SAA oligomers in vitro. It seems worthwhile to address this contrast. For example, the marginal stability of apoSAA oligomers may make them just stable enough to not aggregate in vivo but primed for interacting with lipids to form stable complexes like the ones shown in this study.

10) Typographical errors and formatting requests:

In Figure 5B, please label the heating and cooling curves clearly.

The following sentence has a grammar issue: "…while SAA preferentially binds to such highly curved surfaces of forms them de novo by solubilizing lipid bilayers (Jayaraman et al., 2018; Lu et al., 2014)." It seems like "of forms" should be "or forms".

11) Do the authors mean "oligomers" or "aggregates" or are they being used interchangeably? For example: "hSAA1.1 is less water-soluble in lipid-free state and forms larger ~9 nm aggregates seen on the non-denaturing PAGE (hSAA, Figure 6—figure supplement 2A) as compared to ~7.5 nm aggregates formed by free mSAA1.1". Later in the Discussion (third paragraph), there is mention of SAA oligomers.

[Editors' note: further revisions were requested prior to acceptance, as described below.]

Thank you for resubmitting your work entitled "Synergy between serum amyloid A and secretory phospholipase A_2_ suggests a key role of ancient protein in lipid clearance" for further consideration at *eLife*. Your revised article has been favorably evaluated by Vivek Malhotra as the Senior Editor and a Reviewing Editor.

The manuscript has been improved but there are some remaining issues that need to be addressed before acceptance, as outlined below:

All the technical points have been adequately addressed. However, a major concern lingers about the relevance of these findings in a biological setting. The authors do point out this shortcoming in their rebuttal letter – "In our view, a more detailed discussion of these and other functional properties of SAA-only particles compared to classical lipoproteins is premature until SAA-only particles are found in vivo or in a cell-based system". Please revise the manuscript.

---

## [Author Response]

Essential revisions:1) A major component of the author's model is the pH dependence of this process. However, this point is not rigorously developed (Figure 6 shows that SAA can remove FFA at acidic pH, but it does not show and pH dependence of SAA). Further examination of the pH dependence of SAA dependent remodeling of MLV would strengthen their hypothesis.

The pH dependence of the lipid remodeling by SAA is, indeed, an important point that has been explored in detail in our previous work (Jayaraman et al., 2017). This point is now briefly addressed in the revised Results as follows:

“Previously we showed that the SAA structure, stability and ability to remodel POPC MLV remain invariant at pH 7.5-5.5 but are altered at near-lysosomal pH (Jayaraman et al., 2017). Here, we explored how pH influences the ability of SAA to remove FFA from human lipoproteins.”

2) Are the 7 nm particles micelles or lamellar structures? EM structural analysis would be very informative. It is not clear why the 7 nm particle would be a better vehicle for retrieving cell debris than either an emulsion like LDL/vLDL/HDL or lamellar HDL. Addressing this by experiments or in the Discussion would increase the impact of their model.

Firstly, the 7 nm particles must be micellar since they contain FFA and lysoPC, which are micelle- and bicelle-forming lipids. Unfortunately, the EM analysis of these particles was not as informative as suggested by the reviewer. This was due, in part, to the small particle size (similarly, 7-8 nm HDL are also not well-resolved by negative-stain EM). The problem was confounded by the particle aggregation and by our empirical observation that SAA does not stain quite as well as other apolipopoteins. A detailed structural analysis of these particles is subject of our ongoing studies by other methods and will be described elsewhere

Secondly, we emphasize that the 7 nm SAA-only particle is not a classical lipoprotein but a protein-rich complex. We are not claiming that such 7 nm SAA-containing particles are better at lipid removal than HDL, LDL, VLDL. However, as stated throughout the manuscript (e.g. Discussion), SAA has several advantages over larger apolipoproteins found on HDL, LDL and VLDL:

i) SAA is dramatically upregulated in inflammation, both systemically and locally at the sites of injury; in contrast, plasma levels of other apolipoproteins and albumin drop. Therefore, lipid removal by SAA in inflammation is governed, in part, by the mass action law;

ii) As shown in Figure 1A, SAA spontaneously solubilizes lipids in an energy-independent process. In contrast, formation of HDL, LDL and VLDL requires a complex molecular machinery and ATP hydrolysis, which does not function in dead cells at the sites of inflammation and injury (please see the Discussion).

Moreover, internalization of SAA-only lipoproteins by cells proceeds via different receptors and different biochemical mechanisms than selective uptake of HDL lipids by hepatocytes via SR-BI receptor. We also note that LDL and VLDL do not remove lipids from peripheral cells in vivo. In our view, a more detailed discussion of these and other functional properties of SAA-only particles compared to classical lipoproteins is premature until SAA-only particles are found in vivo or in a cell-based system.

*3) Based on the large particle size, what would be the oligomeric state of mouse and human SAA in these structures, how does it compare to the oligomeric structures seen for mouse and human SAA* in vitro*? Are these potentially the oligomeric structures alluded to in the following statement: "Unlike albumin, SAA forms oligomers to sequester lipids. Although the structure of these oligomers is unknown, lipids are expected to bind in a hydrophobic cavity formed by concave apolar faces of amphipathic helices from several protein molecules (Frame and Gursky, 2016; Frame et al., 2017)."*

In an attempt to determine the oligomeric state of SAA in lipid complexes, we have now included cross-linking data of SAA-POPC complexes in new panels D, E in Figure 4—figure supplement 1. The revised Results section reads:

“To assess the number of SAA molecules per model particle, SAA-POPC were cross-linked with glutaraldehyde. SDS PAGE of intact SAA-POPC showed sharp bands corresponding to protein monomer, dimer and trimer, while hydrolyzed SAA-POPC showed a prominent hexamer band (Figure 4—figure supplement 1D), suggesting that each particle contained at least six protein molecules.”

Based on the published data on SAA1.1 and SAA2 work from Dr. Colon’s group (e.g. PMID: 21439938, 24706838 and references therein), the oligomeric form of SAA in 7 nm particle resembles the hexamer found in solution and in one crystal structure (PMID: 24706838). However, please note that the results of cross-linking alone are insufficient to determine the exact oligomeric state of the protein. Therefore, we opted not to discuss this point in detail in the revised manuscript.

4) The authors state that MLVs are not hydrolyzed by sPLA_2_ in the absence of SAA. Data should be provided to support this statement.

We agree and provide such data in Figure 1 where a new panel D has been added in revision. Revised Figure 1 legend reads:

“(D) Thin-layer chromatography analysis of POPC MLV before (intact) or after their incubation with sPLA_2_-III (+ sPLA_2_-III). The PC band is indicated; the absence of the lyso-PC band underneath the PC band indicates the absence of significant hydrolysis.”

Revised Results section reads:

“In contrast, MLV were not hydrolyzed by sPLA_2_: the levels of FFA were below the detection limit of our assay, and thin-layer chromatography showed only the presence of PC (Figure 1D).”

5) The authors state that PLA_2_ incubation with SUVs leads to hydrolysis of 40-50% of POPC, how does this compare to SAA-treated MLVs in terms of percentage. In other words, in Figure 1C, what percentage of total oleic acid does that represent?

This point is now addressed in revised legend to Figure 1C as follows:

“In SAA-POPC particles, approximately 70% of the total lipid was hydrolyzed by sPLA_2_-III and 60% by sPLA_2_-IIa.

6) Between the first and second section of the Results the authors switch from MLVs to SUVs. This should be discussed or justified. Otherwise it is difficult to relate the first portion of the manuscript to the subsequent analysis.

We agree and address this point in the revised Results:

“The ~8 nm SAA-POPC complexes formed upon spontaneous solubilization of MLV using 1:10 to 1:100 protein:lipid molar ratio were nearly invariant in size (Figure 1C). […] Therefore, in the current study, we used POPC SUV to test the effect of the particle size on the lipolysis by sPLA_2_.”

7) The studies of isolated HDL and LDL would be clarified by comparison to model membranes that more closely resemble the particles, for example ApoA-1 mimetic nano-particles and emulsion particles.

To the best of our knowledge, SAA in vivo associates with mature lipoproteins, mainly HDL but also LDL and VLDL, but not with nascent “discoidal” HDL such as nanoparticles containing apoA-I or apoA-I mimetics. Therefore, comparison with such nanoparticles would not add physiologic relevance to our work. Similarly, we believe that using emulsion-like model particles does not provide any additional advantage over human plasma lipoproteins.

Please note that previously we and others showed that SAA can solubilize a wide range of lipids varying in molecular shape and charge, e.g. PCs, PE, PG, lysoPC, FFA, cholesterol and their mixtures (Jayaraman et al., 2017 and references therein). Therefore, as stated in Jayaraman et al., 2018, and in the current manuscript, we propose that SAA provides a scavenger for a wide range of lipids regardless of their exact composition.

8) The blots in Figure 3 are difficult to see. Please provide a longer exposure, or increased contrast?

Please note that Figure 3 shows native PAGE stained with a protein stain; these are not blots. The contrast in this figure is typical of that seen in our other native gels, at least on our computer. To comply with the strict journal’s policy on image presentation, we opted not to manipulate the original data. However, if the manuscript is accepted, we will work with the editorial office to modify the figures and improve the contrast according to their instructions.

*9) The formation of hyperstable complexes between SAA and lipid hydrolysis products is opposite to the known marginal stability of SAA oligomers* in vitro*. It seems worthwhile to address this contrast. For example, the marginal stability of apoSAA oligomers may make them just stable enough to not aggregate* in vivo *but primed for interacting with lipids to form stable complexes like the ones shown in this study.*

We agree with this interpretation and include it in the Discussion:

“High stability of SAA complexes with lipids and their hydrolytic products contrasts with the marginal in vitro stability of lipid-free SAA oligomers above pH 5 ((Jayaraman et al., 2017) and references therein). We speculate that these marginally stabile protein oligomers are primed for interacting with lipids to form stable complexes like those reported in the current study.”

10) Typographical errors and formatting requests:In Figure 5B, please label the heating and cooling curves clearly.

*The following sentence has a grammar issue: "…while SAA preferentially binds to such highly curved surfaces of forms them* de novo *by solubilizing lipid bilayers (Jayaraman et al., 2018; Lu et al., 2014)." It seems like "of forms" should be "or forms".*

Thank you – the heating and cooling data in the figures have been clearly labeled and the typo in the text has been corrected in revision.

11) Do the authors mean "oligomers" or "aggregates" or are they being used interchangeably? For example: "hSAA1.1 is less water-soluble in lipid-free state and forms larger ~9 nm aggregates seen on the non-denaturing PAGE (hSAA, Figure 6—figure supplement 2A) as compared to ~7.5 nm aggregates formed by free mSAA1.1". Later in the Discussion (third paragraph), there is mention of SAA oligomers.

In the revised manuscript, the term “aggregates” is replaced with “oligomers” to avoid confusion.